# Position error-free control of magnetic domain-wall devices via spin-orbit torque modulation

Seong-Hyub Lee [1,3], Myeonghoe Kim [1,3], Hyun-Seok Whang[1,3], Yune-Seok Nam[1], Jung-Hyun Park[1], Kitae Kim[1], Minhwan Kim[1,2], Jiho Shin[1], Ji-Sung Yu [1], Jaesung Yoon [1], Jun-Young Chang[1,2], Duck-Ho Kim [2] & Sug-Bong Choe[1] ✉

Magnetic domain-wall devices such as racetrack memory and domain-wall shift registers facilitate massive data storage as hard disk drives with low power portability as flash memory devices. The key issue to be addressed is how perfectly the domain-wall motion can be controlled without deformation, as it can replace the mechanical motion of hard disk drives. However, such domain-wall motion in real media is subject to the stochasticity of thermal agitation with quenched disorders, resulting in severe deformations with pinning and tilting. To sort out the problem, we propose and demonstrate a new concept of domain-wall control with a position error-free scheme. The primary idea involves spatial modulation of the spin-orbit torque along nanotrack devices, where the boundary of modulation possesses broken inversion symmetry. In this work, by showing the unidirectional motion of domain wall with position-error free manner, we provide an important missing piece in magnetic domain-wall device development.

In magnetic domain-wall devices[1–6], data bits are stored in a sequence of magnetic domains along nanowire tracks. Then, with the movement of such magnetic domains along the tracks, stored data bits can be accessed sequentially at the read-and-write junctions. This data access scheme is basically the same as that of hard disk drives[7], which store data bits in a sequence of magnetic domains along azimuthal tracks on circular disks. With the rotation of these circular disks, stored data bits can be accessed by read-and-write heads. Instead of such mechanical rotation in hard disk drives, magnetic domain-wall devices utilize current-induced magnetic domain-wall motion[1,3,4,8–16] on mechanically fixed tracks. Thus, magnetic domain-wall devices offer better stability and lower power consumption than hard disk drives while maintaining massive data storage capabilities. Therefore, extensive efforts have been devoted toward the development of better-performing magnetic

domain-wall devices with fast operation speeds[10,12] and low power consumption[13].

Controlling domain-wall motion as perfectly as mechanical motion is the most critical issue in magnetic domain-wall devices. However, unfortunately, domain-wall motion exhibits current-induced tilting[14–17] and stochastic pinning at quenched disorders in real media[18–22]. The current-induced Oersted field is known to tilt domain walls[14,15]. Fairly recently, researchers have discovered that the Dzyaloshinskii–Moriya interaction[23–25] also exerts large tilting torques on domain walls in chiral magnetic materials[16,17]. Here, the tilting directions are opposite between up–down and down–up domain walls. Thus, such tilting causes collision and collapse between adjacent up–down and down–up domain walls. Synthetic antiferromagnetic layers[26] have been adopted to avoid tilting by compensating for the

[1]Department of Physics and Astronomy, Seoul National University, Seoul 08826, Republic of Korea. [2]Center for Spintronics, Korea Institute of Science and Technology (KIST), Seoul 02792, Republic of Korea. [3]These authors contributed equally: Seong-Hyub Lee, Myeonghoe Kim, Hyun-Seok Whang. ✉e-mail: sugbong@snu.ac.kr

tilting torques between antiferromagnetically aligned magnetic layers. However, even with such synthetic antiferromagnetic layers, tilting and deformation are induced by the stochastic pinning and depinning processes with quenched disorders[22], and such quenched disorders are inevitably generated during the deposition of real media.

Owing to the stochastic nature, domain walls randomly stop at unpredictable locations, causing position errors. Several attempts have been made to address this stochastic nature by adopting artificial notches[1,27–30], additional layered structures[31], and local modification of magnetic properties[32,33]. However, such artificial structures create an additional pinning potential that pins the domain walls inside. A stronger pinning potential is desirable for more precise position control of domain walls. However, a stronger pinning potential also requires a higher electric current to depin from the potential. Thus, a dilemma exists in determining the pin-

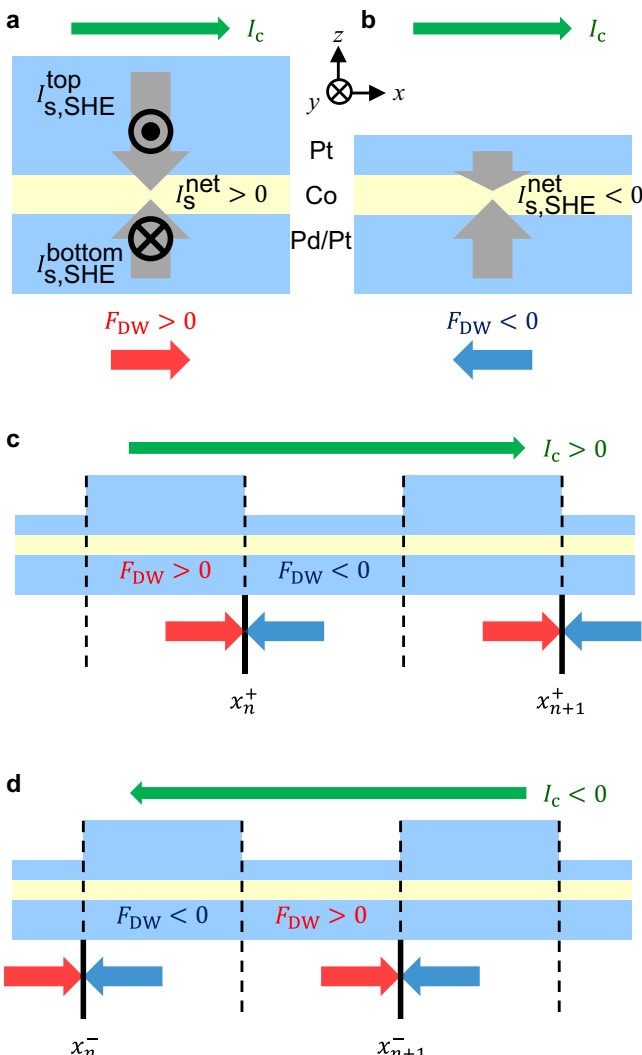

**Fig. 1 | Modulation of the spin-orbit torque by adjusting Pt layer thickness.**
**a**, **b** Opposite $I_{\text{s,SHE}}^{\text{net}}$ and opposite $F_{\text{DW}}$ in Pt/Pd/Co/Pt films with thicker **a** and thinner **b** top Pt layers. The green horizontal arrows on the top show the direction of the injected electric current. The gray vertical arrows show $I_{\text{s,SHE}}^{\text{top}}$ and $I_{\text{s,SHE}}^{\text{bottom}}$ from the top and bottom layers, respectively. The black symbols ⊙ and ⊗ show the direction of spins in the injected spin current. The red and blue horizontal arrows on the bottom show the direction of $F_{\text{DW}}$. **c**, **d** Domain-wall locking at the modulation boundary $x_n^+$ **c** and $x_n^-$ **d** owing to the compression forces (red and blue arrows) under the influence of a positive and negative current, respectively.

ning potential strength to achieve precise position control and low power consumption. As optimizing solely for precise position control is not possible, such compromised pinning structures are inevitably accompanied by position errors. Several correction software technologies are actively being explored[34,35] to address these inevitable position errors.

In this work, we propose a new concept of domain-wall control with a position error-free and tilting error-free scheme. In this scheme, the sign of the spin-orbit torque[8,36] is modulated spatially along nanowires. Notably, opposite signs of the spin-orbit torque generate opposite driving forces on domain walls. Therefore, applying an electric current with an appropriate polarity pushes the domain wall toward the modulation boundary from both sides. Thus, the domain wall is locked at the modulation boundary. With this, the domain wall cannot jump to the other neighboring position, as the main driving forces keep compressing the domain wall toward the modulation boundary. Moreover, the tilting angle has to be reset to the angle of the modulation boundary, despite the occurrence of tilting during domain-wall motion. Therefore, spin-orbit torque modulation enables position error-free and tilting error-free control of domain walls. Unlocking the domain wall can be accomplished by reversing the polarity of the electric current. An asymmetric geometry of the modulation boundary induces an asymmetric and thus, unidirectional unlocking of domain walls. By repeating these procedures, domain walls can keep moving in a unidirectional manner by one data bit per clock pulse of the alternating electric current, as confirmed by micromagnetic simulations. Moreover, the position error-free control of domain walls is also demonstrated experimentally. In the experiments, devices with periodic modulation of the spin-orbit torque are fabricated with a 250 nm modulation. These devices exhibit position error-free operation up to eight bits within a detecting laser spot of 2 μm in diameter.

## Results

### Domain-wall locking by spin-orbit torque modulation

The sign and magnitude of the spin-orbit torque can be controlled by adjusting the thicknesses of heavy metal layers adjacent to the ferromagnetic layer[8,36]. Herein, we prepared Pt/Pd/Co/Pt films by modulating the top Pt layer thickness to induce sign reversal of the spin-orbit torque while preserving the Dzyaloshinskii–Moriya interaction (See Methods for details). As depicted in Fig. 1a, when an electric current $I_c$ (green arrow) flows through the Pt/Pd/Co/Pt films, the spin Hall effect[37,38] at the top Pt and bottom Pt/Pd layers injects spin currents $I_{\text{s,SHE}}^{\text{top}}$ and $I_{\text{s,SHE}}^{\text{bottom}}$ (gray arrows) into the Co layer, respectively. The direction of the injected spins is determined by the cross product between the directions of $I_c$ and $I_{\text{s,SHE}}$. For the present geometry, the spins $\sigma_y$ along the $\pm y$ direction are injected, with $I_c$ along the $+x$ direction and $I_{\text{s,SHE}}$ along the $\pm z$ direction. These injected spins have opposite polarity (black symbols) between the top and bottom layers. Therefore, the net amount of the injected spin current $I_{\text{s,SHE}}^{\text{net}}$ is determined based on the counterbalance between the injected spin currents i.e., $I_{\text{s,SHE}}^{\text{net}} = I_{\text{s,SHE}}^{\text{top}} - I_{\text{s,SHE}}^{\text{bottom}}$. As a thicker layer generates a larger spin current within the range of the spin diffusion length[39–41], the sign of $I_{\text{s,SHE}}^{\text{net}}$ can be reversed by adjusting the thicknesses of the top Pt layers, as depicted in Fig. 1b. Consequently, the sign of the spin-orbit torque can also be reversed in the same sense because the spin Hall effect is one of the major sources of the spin-orbit torque[8,36,41–43]. The spin-orbit torque generates the effective magnetic field $H_{z,\text{eff}}$ to domain walls through the relation $H_{z,\text{eff}} \propto I_{\text{s,SHE}}^{\text{net}}(m_x \times \sigma_y)$, where $m_x$ is the $x$ component of the magnetization inside the domain walls. For efficient spin-orbit torque effect, the Nèel-type domain-wall

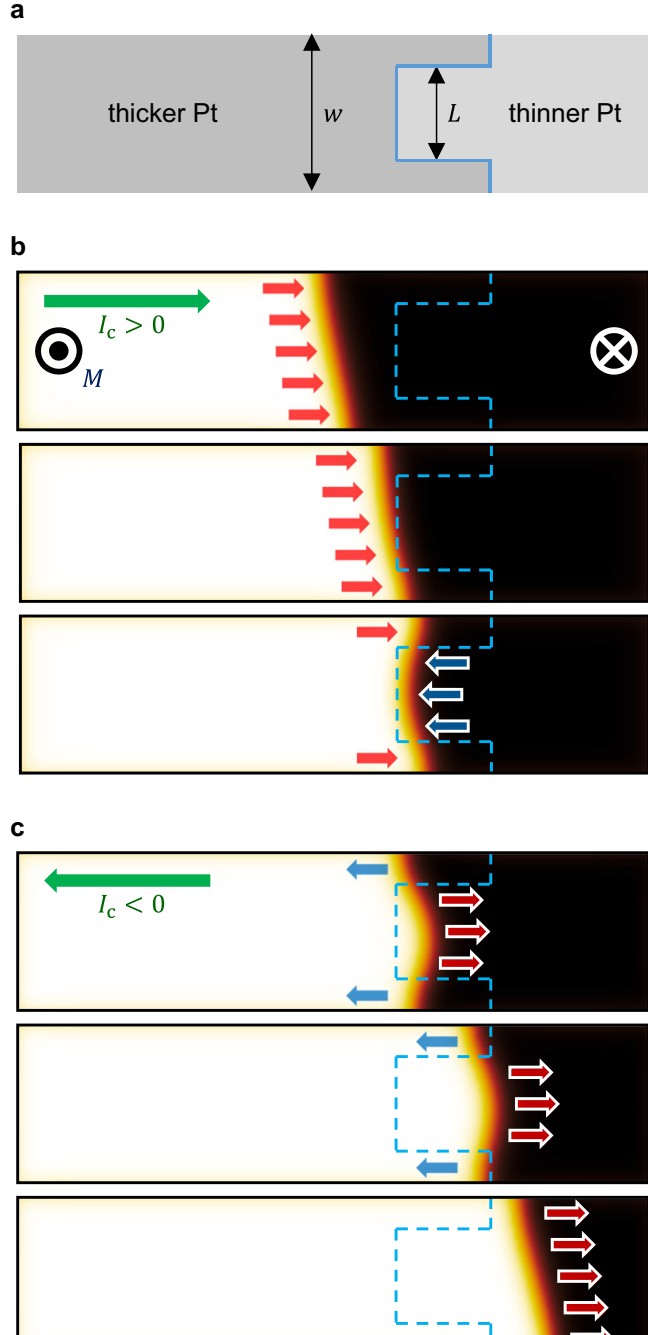

**Fig. 2 | Unidirectional unlocking of domain walls. a** Typical geometry of the asymmetric modulation boundary (solid blue line) between the thicker (dark gray) and thinner (light gray) Pt areas, where $w$ and $L$ are the total wire width and central-area width, respectively. **b**, **c** Sequential snapshot images of the micromagnetic simulation for the situations of domain-wall locking **b** and unlocking **c** owing to the positive and negative directions (green horizontal arrows) of injected current, respectively. White and black areas correspond to the domains magnetized either out of (⊙) or into (⊗) the plane, respectively. Red and blue arrows show the directions of the domain-wall driving force.

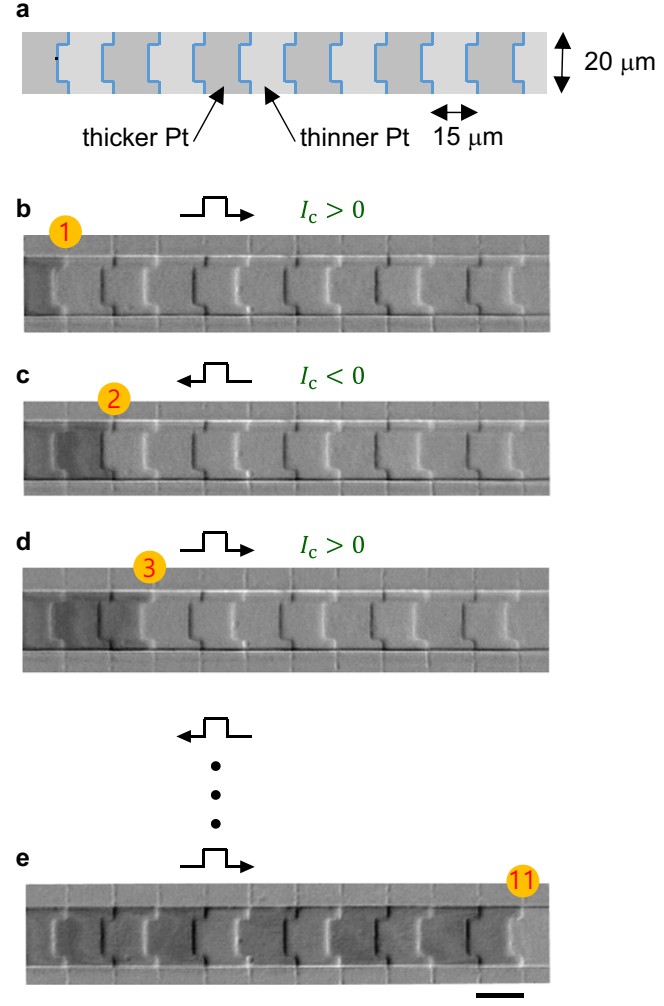

**Fig. 3 | Position error-free control of domain-wall in microdevice. a** Schematic geometry of a microdevice with 20 µm width and 15 µm per bit. **b**–**e** MOKE images of error-free domain-wall position shift and locking at modulation boundaries for each alternating clock pulse.

force $F_{\mathrm{DW}}$. The direction of $F_{\mathrm{DW}}$ is determined by the direction of $H_{z,\mathrm{eff}}$ and consequently, opposite $F_{\mathrm{DW}}$'s are generated by reversing $I_{\mathrm{S,SHE}}^{\mathrm{net}}$, as shown by the red and blue arrows in Fig. 1a, b.

The proposed layer-thickness adjustment can be applied to continuous magnetic wire structures using lithography. Figure 1c presents an example of periodic modulation of the top layer thickness. When an electric current is applied to this structure, the direction of $F_{\mathrm{DW}}$ also becomes modulated according to the layer-thickness modulation. Consequently, domain walls are compressed from both sides and thus, locked at the positions $x_n^+$ (solid vertical lines) of the modulation boundaries, as depicted in Fig. 1c. It is worthwhile to note that this compression is attributed to the major driving force on the domain walls. Therefore, the compression force is meaningfully stronger than the pinning force because the driving force must be stronger than the pinning force. Otherwise the device would not function. Moreover, the area of the compression force is significantly wider than the area of the pinning force. The compression force is applied to the whole area over one modulation period (i.e., the size of two data bits). By contrast, the pinning force appears adjacent to the pinning site, which has to be much smaller than the size of one data bit. Therefore, position errors

configuration ($|m_x| = 1$) is considered with a sizeable Dzyaloshinskii−Moriya interaction (See Methods for experimental realization). Then, under application of non-zero $H_{z,\mathrm{eff}}$ along the $z$ axis, the domain wall moves along the wire (i.e., the $\pm x$ direction), which can be interpreted as the motion under the corresponding driving

do not likely occur owing to an unwanted position jump in overcoming the strong compression force over the wide area.

By inverting the polarity of $I_c$, the domain wall can be unlocked from positions $x_n^+$ and then pushed to the other alternating positions $x_n^-$, as shown in Fig. 1d. The direction of unlocking, either right or left, can be determined based on the geometry of the modulation boundary. Basically, the broken left-to-right inversion symmetry of the modulation boundary provides an unequal chance of unlocking either to the right or to the left. Based on this concept, several peculiar geometries of broken inversion symmetry are available for the induction of unidirectional unlocking.

### Micromagnetic simulation of unidirectional unlocking

One of the peculiar geometries is presented in Fig. 2a. The key features here are the unequal widths between the central (light gray) and outer (dark gray) areas of different Pt thicknesses. Figure 2a presents a case of $L > w/2$, where $w$ is the wire width, and $L$ is the width of the central area (for a case of $L < w/2$, see Supplementary Note. 1 and Supplementary Fig. 1). Figure 2b illustrates snapshots of a micromagnetic simulation for when a domain wall is pushed toward this boundary, where $w = 100$ nm and $L = 60$ nm (see Methods for simulation details). Here, the black and white areas correspond to the down and up domains, respectively, with the domain wall in between. In the present situation, a positive current ($I_c > 0$) generates a positive driving force (red arrows) that pushes the domain wall to the right. As the domain wall travels across the boundary, the domain wall stops under the reverse force (blue arrows) from the central area of the thickness modulation. It is worth noting that when the domain wall stops at the equilibrium position, the domain wall lies more on the central area owing to its unequal widths. If one reverses the current polarity at this moment, the driving forces are also reversed to push the domain wall away from the modulation boundary. Subsequently, owing to the unequal widths, the domain wall is pushed more toward the central-area side, as shown in Fig. 2c (see Supplementary Movie 1 and 2). Consequently, the domain wall is always released along the direction determined by the peculiar, broken symmetry of the modulation boundary. The unidirectional unlocking direction is the same for the up–down and down–up domain walls and the thicker–thinner and thinner–thicker modulation boundary, that is, always toward the wider-area side.

### Experimental proof of principles based on micro device

The proposed position error-free scheme was then confirmed experimentally on two different scales: (1) microscale devices for proof-of-principles with better visualization and (2) nanoscale devices for practical device-scale scalability. These devices have the same stacking layers Pt/Pd/Co/Pt, grown by DC magnetron sputtering (see Supplementary Note 2 for details). The device structures were made by either optical or electron lithography for the microscale and nanoscale devices, respectively. Figure 3a presents the microscale device structure with periodic $t_{Pt}$ modulation on the wire structure, where each modulation section is 15 μm wide. Here, $w = 20$ μm, and $L = 12$ μm. A domain wall was initially placed at the first modulation boundary, labeled by 1 in Fig. 3b. In the image, the weak periodic contrast can be attributed to the topography of thickness modulation. In addition to the topographical contrast, the additional darker contrast on the leftmost bit corresponds to the reversed domain owing to the magnetooptical Kerr effect (MOKE), different from the lighter contrast on the other bits of the unreversed domain (see Supplementary Note 3 and Supplementary Fig. 3 for details). This domain wall was locked exactly at the modulation boundary by injecting a positive $I_c$, as in the situation shown by Figs. 1c and 2b.

By inverting the polarity of $I_c$ with the same pulse width, the domain wall was unlocked from the first modulation boundary. The domain wall was then pushed to the right unidirectionally under

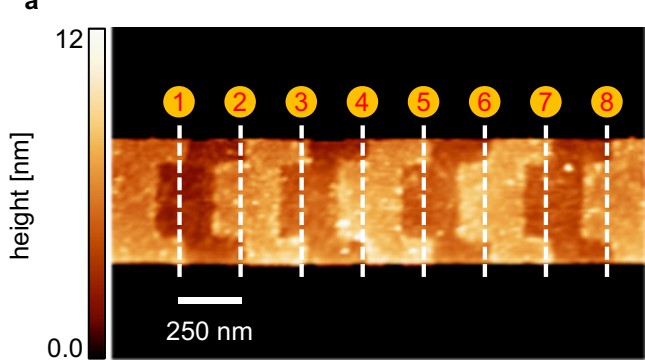

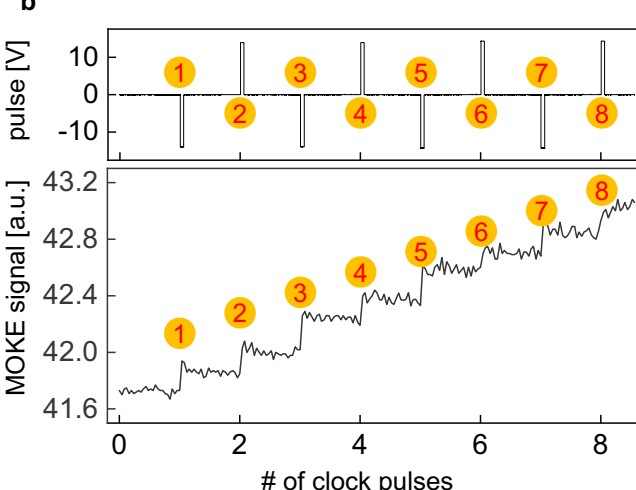

**Fig. 4 | Position error-free control of domain wall in nanotrack. a** Atomic force microscope image of the nanotrack with Pt-thickness modulation of 250 nm per bit. **b** Injected pulse (top panel) and MOKE signal (bottom panel) with respect to the number of alternating clock pulses.

the broken symmetry with unequal widths, as in the situation shown in Fig. 2c. Consequently, the domain wall was locked exactly at the next modulation boundary, labeled by 2 in Fig. 3c. Note that once locked, the domain wall does not move under any further current pulses with the same polarity, regardless of whether they are stronger or weaker in magnitude. However, by inverting the polarity of $I_c$, the domain wall was pushed to the right again and then locked at the next modulation boundary, labeled by 3 in Fig. 3d. By repeating this procedure, the domain wall kept moving to the right by one bit per clock pulse. Figure 3e presents the domain wall locked at the rightmost modulation boundary after injecting several alternation current pulses (see Supplementary Movie 3). The present observation evidently demonstrates the position error-free control of domain walls at each modulation boundary.

### Demonstration of position error-free control in nanotrack

To test whether the present position error-free control scheme also functions in nanoscale devices, we fabricated 250 nm wide modulations on a nanowire structure. Figure 4a presents an atomic force microscope image of the nanotrack device, where $w = 500$ nm and $L = 300$ nm (see Supplementary Note 4 and Supplementary Fig. 4 for details). The present nanotrack structure is the same as the microdevice except for the size. Owing to the size being smaller than the optical diffraction limit, the domain-wall position was monitored by the MOKE signal from a laser spot (~2 μm in diameter) covering the entire device area in the image. Similar to the MOKE images in Fig. 3, as

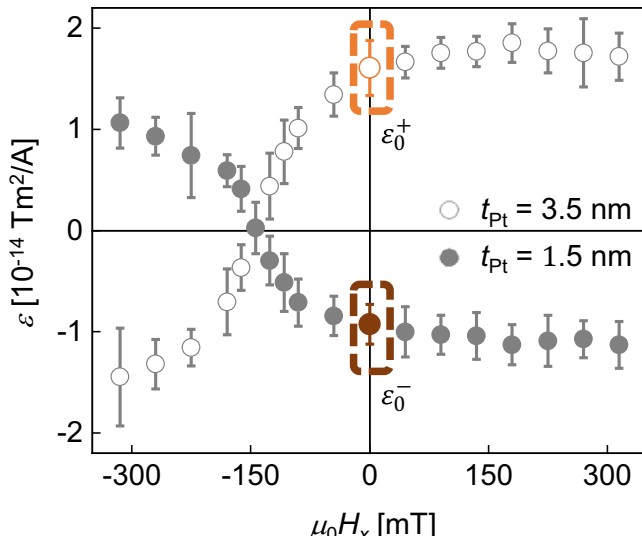

**Fig. 5 | Modulation of $\varepsilon$ with different $t_{Pt}$.** Plot of $\varepsilon$ with respect to $H_x$ for Pt/Pd/Co/Pt films with different $t_{Pt}$, as denoted inside the plot. Dashed boxes highlight the zero-field values $\varepsilon_0^+$ and $\varepsilon_0^-$, which are opposite to each other.

the domain wall moves across the image, the average MOKE intensity of the image also changes accordingly. Thus, the MOKE signal level indicates the position of the domain wall.

A domain wall was initially placed on the leftmost position of the image. The same width (30 ms) of alternating current pulses were then injected, as shown in the top panel of Fig. 4b, while monitoring the MOKE signal from the laser spot on the device area in the image, as shown in the bottom panel. Notably, the stepwise MOKE signal indicates that the domain-wall position can be well controlled by locking at each modulation boundary and shifting one bit per clock. Here, optimized operating conditions were established based on repeated experiments with various pulse amplitude and width sets. Under these optimized conditions, the position error-free scheme was successfully implemented in the nanotrack device.

## Discussion

Here, we propose a novel strategy for controlling the position of domain walls in a position error-free and tilting error-free scheme with spin-orbit torque modulation. The present idea is supported by micromagnetic simulations and validated through experimental investigations in two different device scales by confirming proof-of-principles from microdevices and then successfully extending them to nanotrack devices. The present position-error-free scheme can be integrated with the unidirectional domain-wall control, which is one of the major feasible schemes for magnetic racetrack memory[1]. Our strategy provides an approach to solve an important missing piece in the development of magnetic domain-wall devices.

## Methods

### Spin-orbit torque modulation

Pt/Pd/Co/Pt films were prepared with different $t_{Pt}$ to induce sign reversal of the spin-orbit torque while preserving the Dzyaloshinskii–Moriya interaction. The detailed stacking structure was 5.0 nm Ta/2.5 nm Pt/0.3 nm Pd/0.4 nm Co/$t_{Pt}$ Pt, deposited on 525 μm Si/100 nm SiO₂ substrates by dc magnetron sputtering, where $t_{Pt} = 1.5$ and 3.5 nm. The ultrathin Pd insertion layer induced the Dzyaloshinskii–Moriya interaction by breaking the structural inversion symmetry (see Supplementary Note. 5 and Supplementary Fig. 5 for details)[44–47]. Figure 5 presents the spin-orbit torque efficiency $\varepsilon$ with respect to the in-plane magnetic field $H_x$ for the films (see Supplementary Note 6 and Supplementary Figs. 6 and 7 for details). Owing to

the inversion of the spin Hall effect[8,36] with different $t_{Pt}$, only the sign of $\varepsilon$ was reversed between the films, whereas the overall shape was preserved. The x-axis intersection point was also preserved, corresponding to the effective field induced by the Dzyaloshinskii–Moriya interaction (see Supplementary Note 7 and Supplementary Figs. 8 and 9 for details)[48,49]. In practical device operation, current-induced domain-wall motion can be realized without an external magnetic field, i.e., with the zero-field efficiency $\varepsilon_0^\pm$ highlighted in the plot.

### Micromagnetic simulation

The micromagnetic simulation used the object-oriented micromagnetic framework[50] with the Dzyaloshinskii–Moriya interaction module[51]. Typical Pt/Co/Pt film values were used[52]: a saturation magnetization of 1400 kA/m, exchange stiffness of 31 pJ/m, perpendicular magnetic anisotropy of 1.93 MJ/m³, and Dzyaloshinskii–Moriya interaction strength of 2.0 mJ/m². The cell size was set to 1 nm × 1 nm with 0.3 nm thickness. The simulation size was 100 nm × 400 nm × 0.3 nm. The modulation geometry was given by $L = 0.6w$. The Gilbert damping constant was set to 1.0 for fast stabilization.

## Data availability

The main data supporting the findings of this study are available within this letter and its supplementary information. Extra data can be made available from the corresponding author upon reasonable request. Source data are provided in this paper.

## Code availability

Simulation files have been deposited to the Zenodo public repository and can be accessed via the following link: https://zenodo.org/records/10062835.

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

## Acknowledgements

We thank H.-C. Choi for initial recipes for device fabrication and also PVT (Pure Vacuum Technology) for sputtering chamber maintenance. E-beam lithography is conducted at the Korea Advanced Nano Fab Center (KANC) at the Suwon, Gyeonggi-do, Korea. Photolithography is conducted at institute of applied physics (IAP) at the Seoul National University, Korea. S.H.L., M.K., J.H.P., K.T., Minhwan K., J.S., J.S.Y., J.S.Y., and S.B.C. was supported by the Samsung Science and Technology Foundation (SSTF-BA1802-07), Samsung Electronics Co. Ltd., and the National Research Foundation of Korea (NRF), funded by the Ministry of Science, ICT (MSIT) (2020R1A5A1016518 and 2022M3H4A1A04096339). D.H.K., J.Y.C., and Minhwan K. were supported by the NRF funded by the MSIT (2022R1A2C2004493, 2N70040) and by the Korea Institute of Science and Technology (KIST) institutional program (2E32251).

## Author contributions

S.B.C. proposed the idea for the project and H.S.W. confirmed the idea using the micromagnetic simulation. S.H.L. prepared samples. S.H.L., M.K. and Y.S.N. performed the optical measurements and analyzed the data. J.H.P. and J.S.Y. performed optical measurements. Minhwan K., J.Y.C., D.H.K. and J.H.S. provide resources for sample fabrication. K.K. and J.S.Yoon provide resource for optical and electrical measurement, respectively. S.B.C. and S.H.L. wrote the paper. All authors discussed the results and commented on the paper.

## Competing interests

The authors declare no competing interests.
