## [Peer Review File · Nature Communications]

Position Error-Free Control of Magnetic Domain-Wall Devices via Spin-Orbit Torque ModulationREVIEWER COMMENTS

Reviewer #1 (Remarks to the Author):

The manuscript "Position Error-Free Control of Magnetic Domain-Wall Devices via Spin-Orbit Torque Modulation" written by S.-H. Lee et al. reports clear and interesting results how to control the domain wall position. The paper demonstrated that the spatial modulation of the sign of the spin-orbit torque by modulating the top Pt layer thickness can be a key idea for the position error-free operation of domain wall device. Therefore, I recommend the publication of the manuscript after minor revision for following things.

1. Page 4, line 78 : The authors mentioned that "An asymmetric geometry of the modulation boundary induces an asymmetric and with the rotation of these circular disks thus, unidirectional unlocking of domain walls." This text should be explained in a way that is easy for readers to understand, if possible, with a reference.
2. Page 5, line 96 : The authors argued that a thicker layer generates a larger spin current. Although there is a modulation of Pt layer thickness, the magnitude of the charge current remains constant along the current path, and the charge current density is lower in the thicker layer. The presence of spin-dependent scattering in the thicker Pt layer may be greater, but I don't know about the underlying cause of an increased spin current in a thicker Pt layer. Authors should provide a detailed explanation of the mechanism, if possible, with a reference.
3. Page 6, line 117 : The authors said that the direction of unlocking, either right or left, can be determined based on the geometry of the modulation boundary. Fig.2(a) and 3(a) showed a square-groove shaped boundary along y-axis direction within the plane. It seems that the unidirectional motion of domain wall is dependent on the relative ratio between the total width (W) and the central width (L). It would be helpful if the authors could provide the specific values for W and L . Furthermore, if the authors have obtained results indicating an opposite unidirectional motion in a device with different W and L , it would be beneficial to include those results as supplementary data. I think that some additional discussion on the effect of the ratio L/W or the orientation of the square groove would provide valuable insights.
4. Page 7, line 156 : The authors said that "once locked, the domain wall does not move under any further current pulses with the same polarity." Whether this applies to a larger current pulse or to the same magnitude of current pulse should be clearly mentioned in the text.
5. Figure 4 in Page 24 : It seems that the width of each pulse in Figure 4(a) was varied. The authors should provide information regarding the pulse width in order to clarify how it is used to control the domain wall in Figure 4 and Figure 3.
6. The paper presents very interesting results on the manipulation of domain wall positions. However, I think, there is still a limitation due to the unidirectional motion. Is there a way to achieve bidirectional domain wall motion? Some additional discussion on a potential strategy to overcome the limitation of unidirectional motion would enhance the paper quality.

Reviewer #2 (Remarks to the Author):

The manuscript entitled "Position Error-Free Control of Magnetic Domain-Wall Devices via Spin-Orbit Torque Modulation", by Seong-Hyub Lee, Myeonghoe Kim, Hyun-Seok Whang, Yune-Seok Nam, and Sug-Bong Choe, presents a novel scheme to secure position error-

free control of domain walls for the design of magnetic devices, such as flash memory devices or logic gates. The proper and error-free position control of domain walls is a necessary condition when thinking and engineering the new generation of domain wall based magnetic devices. The proposed solution in the manuscript is of key importance and of major relevance in the field.

As already stated by the authors, several previous works deal with stochasticity and position control of domain walls (Refs. [27-33]), but lacking a definite answer. In this context, the question addressed by this manuscript is not original. However, the authors have found a creative and innovative solution to the problem, which might be of immediate interest for a broad disciplinary audience working on the development of magnetic and related devices.

The results are well supported by the presented data and methodological approach. They used state of the art micromagnetic simulations and report experimental evidence based on two different samples.

Once the main proposed idea is well described and this is supported by both numerical simulations and micro and nanoscale experiments, the conclusions are robust and reliable. Despite this, I have various concerns that have to do mainly with the presentation of the content and the readability of the manuscript. Please consider the next issues as intended to improve the manuscript.

- lines 78-79: The following sentence looks confusing to me: "An asymmetric geometry of the modulation boundary induces an asymmetric and with the rotation of these circular disks thus, unidirectional unlocking of domain walls." What asymmetric is induced? What circular disks? This sentence should be rewritten to gain readability.

- lines 87-101: This paragraph contains the presentation of the key idea of the manuscript: that the counterbalance of spin currents controlled by the top Pt layer thickness, spatially modulated, results in locking at the modulation boundaries, and consequently a position error-free scheme. In my opinion the arguments and presentation on this paragraph can be improved by providing some more details and discussions. The following points should be addressed:

A clear differentiation between charge and spin currents is needed in the notation, since "I" is currently used for both currents. I suggest to use I_c and I_s (or I and J , or other option) for charge and spin currents, respectively.

I think it is advisable to include in the presentation the relationship between the directions of the applied charge current and the resulting spin current and direction of spins, so that it helps to read Fig. 1. What I mean is that it would help to add something like $\langle I \rangle \sim \theta_{SHE} \langle I \rangle_s \times \sigma$, where σ is a unitary vector representing the direction of spins.

The authors state that "... a thicker layer generates a larger spin current ...". Why is that so? As I understand, this is only valid below the spin diffusion length and is due to a simple cumulative effect. Is that correct? Some comment on this issue would be welcome.

Currents are vector quantities, and as such are represented by arrows, which means that the information is contained in the direction and length of the arrow, but not in its width. I find

confusing the use in Fig. 1 of arrows of different widths for the competing spin currents. I suggest to draw arrows of the same width but different lengths.

Given the direction of spins (σ) of induced spin currents as depicted in Fig. 1, I would expect the driving force for the domain wall to be ruled by an effective magnetic field $H_{\text{SOT}} \sim \theta_{\text{SHE}} (\mathbf{m} \times \sigma)$, where \mathbf{m} is the magnetization vector at the center of the domain wall. This effective field then generates the effective forces shown in Fig. 1. What kind of domain wall is considered in Fig. 1? Is it a Bloch domain wall with a magnetization tilting in its center? Notice that if the magnetization points in the same direction as the direction of spins of the spin current, no net SOT on the domain wall is generated.

Instead of referring to the net spin current $I_{\text{SHE}}^{\text{net}}$, I find more pedagogical to directly use the net force on the domain wall; something like $F_{\text{SHE}}^{\text{net}} = F_{\text{SHE}}^{\text{top}} - F_{\text{SHE}}^{\text{bottom}} \sim |H_{\text{SHE}}^{\text{top}}| - |H_{\text{SHE}}^{\text{bottom}}|$. The counterbalance seems more direct when discussing effective forces.

-line 152: The described experimental situation corresponds to a negative charge current I and the locking of the domain wall at the modulation boundary between a thinner Pt layer in the left and a thicker Pt layer in the right. This situation is shown in Fig. 3b. This is linked to the situation shown in Figs. 1d and 2b. However, Fig. 2b refers to a positive charge currents and thicker(thinner) Pt layer in the left(right). I understand that by symmetry these situations are connected, but the reference to these figures might lead to misunderstandings. Either the figures (maybe Fig. 2) should change or a comment should be included. The same is valid for line 155 when comparing with Fig. 2c but considering a positive charge current.

- I have one concern about the samples growth: are both micro and nanoscale samples growth magnetron sputtering technique? Have they subsequently used electron/optical lithography?

Other minor suggestions:

- lines 27-28: I think there are still many missing pieces to secure magnetic domain-wall devices. I would not say this is the "final missing puzzle" but I do agree this is an "important missing piece".

- line 83-84: The use of "duration" might be confusing when referring to space. I suggest to use "250 nm modulation" instead.

- Fig. 1: The pink arrows in a) are hardly visible by colorblind people. Please check figures to consider accessible palettes for colorblind people.

Reviewer: Sebastian Bustingorry

Response to Reviewer 1

< Preface >

The manuscript “Position Error-Free Control of Magnetic Domain-Wall Devices via Spin-Orbit Torque Modulation” written by S.-H. Lee et al. reports clear and interesting results how to control the domain wall motion. The paper demonstrated that the spatial modulation of the sign of the spin-orbit torque by modulating the top Pt layer thickness can be a key idea for the position error-free operation of domain wall device. Therefore, I recommend the publication of the manuscript after minor revision for following things.

Reply:

We appreciate the reviewer for spending substantial amount of time for several helpful suggestions for our manuscript. Thanks to the Reviewer’s valuable comments and suggestions, our present manuscript is improved comparing to the previous version. The detailed responses to the specific comments are as follow.

< Comment 1 >

Page 4, line 78 : The authors mentioned that “An asymmetric geometry of the modulation boundary induces an asymmetric and with the rotation of these circular disks thus, unidirectional unlocking of domain walls.”. This text should be explained in a way that is easy for readers to understand, if possible, with a reference.

Reply:

It appears that an error has occurred during the PDF conversion involving the automated editing tracing process. We revised the sentence by removing the unnecessarily confusing phrase.

List of the revision:

✓ The sentence at the lines 74-75 (page 4) is modified in accordance to the comment.

< Comment 2 >

Page 5, line 96 : The authors argued that a thicker layer generates a larger spin current. Although there is a modulation of Pt layer thickness, the magnitude of the charge current remains constant along the current path, and the charge current density is lower in the thicker layer. The presence

of spin-dependent scattering in the thicker Pt layer may be greater, but I don't know about the underlying cause of an increased spin current in a thicker Pt layer. Authors should provide a detailed explanation of the mechanism, if possible, with a reference.

Reply:

We agree that the total magnitude of the charge current remains constant along the current path. Therefore, the current density should be redistributed for each region of different thickness. Since the conductivity should not be substantially different between the top and bottom Pt layers, the magnitude of the charge current should be larger at the thicker Pt layer. It is well known that the thicker layer generates a larger spin current within the thickness range thinner than spin diffusion length, which is 2~3 nm for Pt [39-41]. Additional information can be found in Supplementary Note 6 and 7, which has been provided by new authors (J.H.P, Minhwan K., J.H.S., J.Y.C., and D.H.K).

List of the revision:

✓ The present discussion is added at the lines 95-96 (page 5) with additional references [39, 40] for the detailed explanation of the mechanism.

< Comment 3 >

Page 6, line 117 : The authors said that the direction of unlocking, either right or left, can be determined based on the geometry of the modulation boundary. Fig.2(a) and 3(a) showed a square-groove shaped boundary along y-axis direction within the plane. It seems that the unidirectional motion of domain wall is dependent on the relative ratio between the total width (W) and the central width (L). It would be helpful if the authors could provide the specific values for W and L. Furthermore, if the authors have obtained results indicating an opposite unidirectional motion in a device with different W and L, it would be beneficial to include those results as supplementary data. I think that some additional discussion on the effect of the ratio L/W or the orientation of the square groove would provide valuable insights.

Reply:

We appreciate the reviewer for the present suggestion. In accordance to the suggestion, we prepared the supplementary information, which include the discussion of the effect of the ratio L/W with the micromagnetic simulation results of an opposite unidirectional motion. Also, we added the description of the detailed geometry in the text.

List of the revision:

✓ The supplementary information is prepared. The values of w and L are specified at the line 136 (page 6), the line 157-158 (page 7), and the line 181 (page 8). Additional information can be found in Supplementary Note 1, Supplementary Movie 1 and 2.

< Comment 4 >

Page 7, line 156 : The authors said that “once locked, the domain wall does not move under any further current pulses with the same polarity.”. Whether this applies to a larger current pulse or to the same magnitude of current pulse should be clearly mentioned in the text.

Reply:

The domain wall does not move regardless of whether they are stronger or weaker in magnitude. We revised the manuscript by specifying this aspect.

List of the revision:

✓ The present discussion is added at the lines 171 (page 8).

< Comment 5 >

Figure 4 in Page 24 : It seems that the width of each pulse in Figure 4(a) was varied. The authors should provide information regarding the pulse width in order to clarify how it is used to control the domain wall in Figure 4 and Figure 3.

Reply:

We appreciate the reviewer for the present suggestion. We used the same pulse width. Thanks to the reviewer's comment, we recognized that the image had inadequate pixel resolution and we revised it with higher pixel resolution.

List of the revision:

- ✓ The same pulse width was specified at the line 166 (page 8) and lines 188-189 (page 9).
- ✓ Figure 4(b) is revised with the higher pixel resolution.

< Comment 6 >

The paper presents very interesting results on the manipulation of domain wall positions. However, I think, there is still a limitation due to the unidirectional motion. Is there a way to achieve bidirectional domain wall motion? Some additional discussion on a potential strategy to overcome the limitation of unidirectional motion would enhance the paper quality.

Reply:

The magnetic racetrack memory can be achieved by either the unidirectional or bidirectional domain-wall control [1]. The present position-error-free scheme can be integrated with the unidirectional control. We appreciate the reviewer for the suggestion of bidirectional domain-wall control, which might provide other valuable opportunity. We revised the manuscript with the discussion of the possible opportunities in practical development of the magnetic racetrack memory.

List of the revision:

- ✓ The present discussion is added at the lines 201-204 (page 9).

Response to Reviewer 2

< Preface >

The manuscript entitled "Position Error-Free Control of Magnetic Domain-Wall Devices via Spin-Orbit Torque Modulation", by Seong-Hyub Lee, Myeonghoe Kim, Hyun-Seok Whang, Yune-Seok Nam, and Sug-Bong Choe, presents a novel scheme to secure position error-free control of domain walls for the design of magnetic devices, such as flash memory devices or logic gates. The proper and error-free position control of domain walls is a necessary condition when thinking and engineering the new generation of domain wall based magnetic devices. The proposed solution in the manuscript is of key importance and of major relevance in the field.

As already stated by the authors, several previous works deal with stochasticity and position control of domain walls (Refs. [27-33]), but lacking a definite answer. In this context, the question addressed by this manuscript is not original. However, the authors have found a creative and innovative solution to the problem, which might be of immediate interest for a broad disciplinary audience working on the development of magnetic and related devices.

The results are well supported by the presented data and methodological approach. They used state of the art micromagnetic simulations and report experimental evidence based on two different samples.

Once the main proposed idea is well described and this is supported by both numerical simulations and micro and nanoscale experiments, the conclusions are robust and reliable. Despite this, I have various concerns that have to do mainly with the presentation of the content and the readability of the manuscript. Please consider the next issues as intended to improve the manuscript.

Reply:

We appreciate the reviewer for spending substantial amount of time for several helpful suggestions for our manuscript. Thanks to the Reviewer's valuable comments and suggestions, our present manuscript is improved comparing to the previous version. The detailed responses

to the specific comments are as follow. Also reference number indicated in this article is based on the revised manuscript.

< Comment 1 >

lines 78-79: The following sentence looks confusing to me:"An asymmetric geometry of the modulation boundary induces an asymmetric and with the rotation of these circular disks thus, unidirectional unlocking of domain walls." What asymmetric is induced? What circular disks? This sentence should be rewritten to gain readability.

Reply:

It appears that an error has occurred during the PDF conversion involving the automated editing tracing process. We revised the sentence by removing the unnecessarily confusing phrase.

List of the revision:

✓ The sentence at the lines 74-75 (page 4) is modified in accordance to the comment.

< Comment 2 >

lines 87-101: This paragraph contains the presentation of the key idea of the manuscript: that the counterbalance of spin currents controlled by the top Pt layer thickness, spatially modulated, results in locking at the modulation boundaries, and consequently a position error-free scheme. In my opinion the arguments and presentation on this paragraph can be improved by providing some more details and discussions. The following points should be addressed:

< Comment 2-1 >

A clear differentiation between charge an spin currents is needed in the notation, since "I" is currently used for both currents. I suggest to use I_c and I_s (or I and J, or other option) for charge and spin currents, respectively.

Reply:

We appreciate the reviewer for the present comment. In accordance to the comment, we revised the present manuscript with the charge current I_c and the spin current $I_{s,SHE}$.

List of the revision:

✓ The notation was revised the lines 88-97 (pages 4-5), line 123 (page 6), line 138 (page 6), line 164, 166 (page 8) and line 172 (page 8).

✓ Figures 1, 2, and 3 with their captions are modified.

< Comment 2-2 >

I think it is advisable to include in the presentation the relationship between the directions of the applied charge current and the resulting spin current and direction of spins, so that it helps to read Fig. 1. What I mean is that it would help to add something like $I \sim \Theta_{\text{SHE}} (I_{\text{s,SHE}} \times \sigma)$, where σ is a unitary vector representing the direction of spins.

Reply:

In accordance to the suggestion, we revised the present manuscript by adding the suggested relation between the direction of injected spins and the directions of I_c and $I_{\text{s,SHE}}$.

List of the revision:

✓ The present discussion is added at the lines 90-93 (page 4-5).

< Comment 2-3 >

The authors state that "... a thicker layer generates a larger spin current ...". Why is that so? As I understand, this is only valid below the spin diffusion length and is due to a simple cumulative effect. Is that correct? Some comment on this issue would be welcome.

Reply:

We agree that the present statement is only valid below the spin diffusion length with a simple cumulative effect. To avoid any possible ambiguity, we revised the manuscript by specifying the valid thickness range thinner than spin diffusion length (2~3 nm for Pt) with references [39-41]. Additional information can be found in Supplementary Note 6 and 7, which has been provided by new authors (J.H.P, Minhwan K., J.H.S., J.Y.C., and D.H.K).

List of the revision:

✓ The present discussion is added at the lines 95-96 (page 5) with additional references [39, 40] for the detailed explanation of the mechanism.

< Comment 2-4 >

Currents are vector quantities, and as such are represented by arrows, which means that the information is contained in the direction and length of the arrow, but not in its width. I find confusing the use in Fig. 1 of arrows of different widths for the competing spin currents. I suggest to draw arrows of the same width but different lengths.

Reply:

We sincerely appreciate the thoughtful and really helpful suggestion. In accordance to the suggestion, we revised the figure.

List of the revision:

✓ Figure 1 is modified in accordance to the reviewer's suggestion.

< Comment 2-5 >

Given the direction of spins (σ) of induced spin currents as depicted in Fig. 1, I would expect the driving force for the domain wall to be ruled by an effective magnetic field $H_{\text{SOT}} \sim \Theta_{\text{SHE}} (m \times \sigma)$, where m is the magnetization vector at the center of the domain wall. This effective field then generates the effective forces shown in Fig. 1. What kind of domain wall is considered in Fig. 1? Is it a Bloch domain wall with a magnetization tilting in its center? Notice that if the magnetization points in the same direction as the direction of spins of the spin current, no net SOT on the domain wall is generated.

Reply:

I agree that no net SOT is generated for the Bloch-type domain walls. Therefore, we considered the situation with the Néel-type domain walls. As shown by Fig. 5, the present films have a sizeable Dzyaloshinskii-Moriya interaction and thus, exhibit the Néel-type domain walls with the SOT-driven domain-wall motion. To avoid any possible ambiguity, we revised the manuscript by specifying the type of the domain walls under consideration. Additional information can be found in Supplementary Note 5, which has been provided by new author (J.S.Y).

List of the revision:

✓ The present discussion is added at the lines 99-107 (page 5).

< Comment 2-6 >

Instead of referring to the net spin current $I_{\text{SHE}}^{\text{net}}$, I find more pedagogical to directly use the net force on the domain wall; something like $F_{\text{SHE}}^{\text{net}} = F_{\text{SHE}}^{\text{top}} - F_{\text{SHE}}^{\text{bottom}} \sim |H_{\text{SHE}}^{\text{top}}| - |H_{\text{SHE}}^{\text{bottom}}|$. The counterbalance seems more direct when discussing effective forces.

Reply:

In accordance to the comment, we revised the manuscript with the concepts of the effective magnetic field $H_{z,\text{eff}}$ and consequently, the driving force F_{DW} .

List of the revision:

✓ The present discussion is added at the lines 104-107 (page 5).

< Comment 3 >

line 152: The described experimental situation corresponds to a negative charge current I and the locking of the domain wall at the modulation boundary between a thinner Pt layer in the left and a thicker Pt layer in the right. This situation is shown in Fig. 3b. This is linked to the situation shown in Figs. 1d and 2b. However, Fig. 2b refers to a positive charge currents and thicker(thinner) Pt layer in the left(right). I understand that by symmetry these situations are connected, but the reference to these figures might lead to misunderstandings. Either the figures (maybe Fig. 2) should change or a comment should be included. The same is valid for line 155 when comparing with Fig. 2c but considering a positive charge current.

Reply:

We replaced Fig. 3 by another experimental result with the situation that the leftmost modulation boundary is thicker-to-thinner one in accordant to the Figs. 1c and 2b. Therefore, in the revised manuscript, the experimental situation matches exactly the situation under the prediction. Additional information can be found in Supplementary Note 3 and Supplementary Movie 3, which has been provided by new author (K.K).

List of the revision:

✓ Figure 3 is replaced by another experimental result.

< Comment 4 >

I have one concern about the samples growth: are both micro and nanoscale samples growth magnetron sputtering technique? Have they subsequently used electron/optical lithography?

Reply:

Yes. Both samples are grown by using DC magnetron sputtering technique and then, subsequently fabricated by use of optical (for the microscale sample) and electron (for the nanoscale sample) lithography. Additional information can be found in Supplementary Note 2 and 4, which has been provided by new author (Jae-Sung Yoon).

List of the revision:

✓ The present discussion is added at the lines 153-156 (page 7).

< Comment 5 >

lines 27-28: I think there are still many missing pieces to secure magnetic domain-wall devices. I would not say this is the "final missing puzzle" but I do agree this is an "important missing piece".

Reply:

We revised the manuscript in accordance to the present suggestion.

List of the revision:

✓The present manuscript is revised at the line 25-26 (page 1) and line 204-205 (page 9).

< Comment 6 >

line 83-84: The use of "duration" might be confusing when referring to space. I suggest to use "250 nm modulation" instead.

Reply:

We revised the manuscript in accordance to the present suggestion.

List of the revision:

✓The present manuscript is revised at the line 79 (page 4).

< Comment 7 >

Fig. 1: The pink arrows in a) are hardly visible by colorblind people. Please check figures to consider accessible palettes for colorblind people.

Reply:

We again sincerely appreciate the thoughtful suggestion. In accordance to the suggestion, we revised the figure.

List of the revision:

✓ Figures 1(a) are modified in accordance to the suggestion.

REVIEWERS' COMMENTS

Reviewer #1 (Remarks to the Author):

The revised manuscript "Position Error-Free Control of Magnetic Domain-Wall Devices via Spin-Orbit Torque Modulation" written by S.-H. Lee et al. was much improved, and the authors properly replied for the reviewer's comments. Therefore, I think that the revised manuscript can be accepted for the publication. The authors can do an additional revision for the following points.

- 1) The effective field is proportional to the current, but the notation for the current (J) in the supplementary note 6 is different from that (I) in the main manuscript page 5. If there is no other reason, those notations should be the same.
- 2) It will be better if the time width of current pulse is presented.

Reviewer #2 (Remarks to the Author):

Review report

Title: Position Error-Free Control of Magnetic Domain-Wall Devices via Spin-Orbit Torque Modulation

Authors: Seong-Hyub Lee, Myeonghoe Kim, Hyun-Seok Whang, Yune-Seok Nam, Jung-Hyun Park, Kitae Kim, Minhwan Kim, Ji-ho Shin, Ji-Sung Yu, Jae-Sung Yoon, Jun-Young Chang, Duck-Ho Kim, and Sug-Bong Choe

Reviewer: Sebastian Bustingorry

The author have managed to provide an improved version of the manuscript entitled "Position Error-Free Control of Magnetic Domain-Wall Devices via Spin-Orbit Torque Modulation". They have satisfactorily answered all the risen comments whilst gaining readability and clarity. Furthermore, relevant information and comments are treated in the Supplementary Information file.

Overall, the manuscript provides novel and key results regarding position error-free control of domain walls in ferromagnetic tracks, with the potential to strongly impact in the engineer design of domain wall based magnetic device. I recommend the publication of this manuscript.

Response to Reviewer 1

< Preface >

The revised manuscript "Position Error-Free Control of Magnetic Domain-Wall Devices via Spin-Orbit Torque Modulation" written by S.-H. Lee et al. was much improved, and the authors properly replied for the reviewer's comments. Therefore, I think that the revised manuscript can be accepted for the publication. The authors can do an additional revision for the following points.

Reply:

We appreciate the reviewer for spending substantial amount of time for several helpful suggestions for our manuscript. Thanks to the Reviewer's valuable comments and suggestions, our present manuscript is improved comparing to the previous version. The detailed responses to the specific comments are as follow. Also reference number indicated in this article is based on the revised manuscript.

< Comment 1 >

The effective field is proportional to the current, but the notation for the current (J) in the supplementary note 6 is different from that (I) in the main manuscript page 5. If there is no other reason, those notations should be the same.

Reply:

We appreciate the reviewer for the present comment. In accordance to the comment, we revised the notation of current in supplementary note 6.

List of the revision:

- ✓ The notation for current in supplementary note 6 is modified in accordance to the comment.
- ✓ The notation for current in supplementary figure 6 and supplementary figure 7 are modified in accordance to the comment.

< Comment 2 >

It will be better if the time width of current pulse is presented.

Reply:

We appreciate the reviewer for the present comment. In accordance to the comment, we add the width of current pulse.

List of the revision:

- ✓ The sentence at the lines 189-190 (page 9) is modified in accordance to the comment.

Response to Reviewer 2

< Preface >

The author have managed to provide an improved version of the manuscript entitled "Position Error-Free Control of Magnetic Domain-Wall Devices via Spin-Orbit Torque Modulation". They have satisfactorily answered all the risen comments whilst gaining readability and clarity. Furthermore, relevant information and comments are treated in the Supplementary Information file.

Overall, the manuscript provides novel and key results regarding position error-free control of domain walls in ferromagnetic tracks, with the potential to strongly impact in the engineer design of domain wall based magnetic device. I recommend the publication of this manuscript.

Reply:

We appreciate the reviewer for spending substantial amount of time for several helpful suggestions for our manuscript.